# Enhanced disulphide bond stability contributes to the once-weekly profile of insulin icodec

František Hubálek [1,2] ✉, Christian N. Cramer [1,2], Hans Helleberg[1], Eva Johansson [1], Erica Nishimura [1], Gerd Schluckebier [1], Dorte Bjerre Steensgaard[1], Jeppe Sturis [1] & Thomas B. Kjeldsen [1]

Insulin icodec is a once-weekly insulin analogue that has a long half-life of approximately 7 days, making it suitable for once weekly dosing. The Insulin icodec molecule was developed based on the hypothesis that lowering insulin receptor affinity and introducing a strong albumin-binding moiety would result in a long insulin half-life, provided that non-receptor-mediated clearance is diminished. Here, we report an insulin clearance mechanism, resulting in the splitting of insulin molecules into its A-chain and B-chain by a thiol–disulphide exchange reaction. Even though the substitutions in insulin icodec significantly stabilise insulin against such degradation, some free B-chain is observed in plasma samples from minipigs and people with type 2 diabetes. In summary, we identify thiol–disulphide exchange reactions to be an important insulin clearance mechanism and find that stabilising insulin icodec towards this reaction significantly contributes to its long pharmacokinetic/pharmacodynamic profile.

Peptide and protein biopharmaceuticals have proven efficacious in the treatment of a wide variety of chronic diseases[1]. Ideally, such treatments should benefit from infrequent dosing to improve patients' compliance and health-related quality of life. Ultra-long pharmacokinetic profiles, facilitating infrequent administrations, require that the therapeutic molecules are exposed to multiple degradation pathways in patients' blood and interstitial fluid for a period of several weeks. Infrequent administration is then enabled by identifying the relevant degradation pathways and stabilising the molecules against such degradation.

Diabetes is a chronic disease, and insulin treatment is life-saving for all people with type 1 diabetes and many with type 2 diabetes, having unmatched glucose-lowering capability. The native insulin molecule has a very short plasma elimination half-life (hereafter referred to simply as half-life) and is required to be injected frequently. Substantial research efforts have been applied to develop basal insulin analogues to avoid frequent dosing, increase convenience and improve treatment compliance and quality of life[2,3]. In 2021, we reported the development of insulin icodec, a basal insulin analogue with a long half-life that would allow once-weekly dosing (A14E B16H B25H B29Nε-C20 diacid-LγGlu−2xAdo desB30 human insulin)[4–7]. The tight binding of insulin icodec to albumin mediated by fatty diacid derivatisation in conjunction with reduced insulin receptor affinity[4,5] results in a circulating albumin-bound insulin icodec depot that is essentially inactive. Upon slow and continuous dissociation from this albumin-bound depot, insulin icodec activates the insulin receptor and elicits its biological function. These modifications introduced into insulin resulted in a once-weekly pharmacokinetic profile with a half-life of 196 h in humans[5].

---

[1]Novo Nordisk A/S, Maaloev, Denmark. [2]These authors contributed equally: František Hubálek, Christian N. Cramer. ✉e-mail: FHUB@novonordisk.com

As mentioned previously, human insulin has a very short half-life (0.08–0.25 h) after intravenous administration owing to a fast clearance that is primarily mediated via the insulin receptor[8]. Its non-receptor-mediated clearance is presumably limited owing to its short half-life. In contrast, an insulin analogue with an ultra-long pharmacokinetic profile, as illustrated by insulin icodec, will have a significantly longer time to interact with components of blood and interstitial fluid, including proteins, enzymes, glutathione, cysteine and other thiols, potentially leading to insulin degradation. Consequently, such non-receptor-mediated clearance mechanisms might impact the pharmacodynamic potency of ultra-long-acting insulin analogues to a significantly higher degree than for the native insulin molecule. Furthermore, examples of insulin analogs with long pharmacokinetic profiles but low in vivo potency in rats were identified in structure–activity relationship of insulin icodec[4]. Thus, non-receptor-mediated clearance might impact the pharmacodynamic potency of some ultra-long-acting insulin analogues.

Here, we show and characterise a degradation mechanism comprising disulphide bond rearrangements that leads to the splitting of insulin analogues into their A-chain and B-chain and concomitantly impairs both their in vivo glucodynamic potency and their half-life. Amino acid substitutions are introduced to insulin icodec to counteract the effect of insulin chain-splitting in a balanced manner with focus on adequate glucose control and a long pharmacokinetic half-life to enable once-weekly dosing.

## Results

### Insulin stability as a function of redox potential

The three disulphide bonds of insulin have different degrees of solvent exposure; the A7-B7 disulphide bond being most exposed, the A20-B19 disulphide bond partially exposed and the A6-A11 disulphide bond almost completely buried inside the structure (Fig. 1a). Furthermore, considerable flexibility in the structure is observed in the N-termini of the A-chain and B-chain, in the C-terminus of the B-chain and in the 'hinge' region of the B-chain. This flexibility results in an increased solvent exposure of the disulphide bonds in the insulin molecule. A thiol–disulphide exchange reaction between a free thiol group and one of these disulphide bonds can eventually lead to the splitting of insulin into its A-chain and B-chain (Fig. 1b). Cysteine residues in each chain will still be oxidised to disulphide bonds but no interchain disulphide bonds will exist to keep the chains connected. Owing to the very low-insulin plasma concentration, the resulting A-chain and B-chain will be practically irreversible end-products of such a thiol–disulphide exchange reaction. Splitting the insulin molecule into its A-chain and B-chain will result in a complete loss of potency because the chains are no longer able to bind to and activate the insulin receptor. The probability of the thiol–disulphide reaction splitting insulin into its chains is expected to be proportional to the circulating half-life of the molecule. While splitting of insulin via the thiol–disulphide reaction is not expected to account for a major clearance mechanism for human insulin, with its short circulation half-life, it will be likely to play a key role during the development of ultra-long-acting insulin analogues, especially with protraction mainly in a circulating plasma depot such as insulin icodec.

We developed an in vitro assay to compare insulin stability under decreasing redox potential conditions and demonstrated that insulin icodec was significantly more resistant than human insulin to thiol–disulphide exchange reaction-driven degradation (Fig. 1d). The results from the assay also confirmed that the A-chain of insulin, which has two intrachain disulphide bonds, and the B-chain of insulin, which has a single intrachain disulphide bond, are the end-products of such degradation (Fig. 1c).

We did not observe any major mixed disulphide products corresponding to the attachment of glutathione to insulin. This is perhaps not surprising because a free thiol present on the same chain would rapidly react with the glutathionylated disulphide to free glutathione and form an intrachain disulphide bond. This is also in line with previous observations of a very fast (half-life of 15 s) disappearance of mixed disulphides formed between lysozyme and cystine by the addition of low-molecular-weight thiols[9]. In further support of this mechanism, we observed a single form of the B-chain and three isoforms of the A-chain resulting from all possible combinations of the four cysteine residues in the A-chain (Fig. 1c and Supplementary Fig. 2). Thus, our results support the previously published hypothesis suggesting that the rate-limiting step of the thiol–disulphide-based reaction is the free thiol (glutathione) attack of a disulphide bond.

Substitutions in insulin amino acid sequence and/or modification of the insulin molecule will alter the folding stability and flexibility of insulin and thus change the solvent accessibility and reactivity of the disulphide bonds. Therefore, we investigated the effects of the individual substitutions present in the insulin icodec molecule on its redox stability. The results indicated that while substitution of Phe with His at position B25 (B25H) or substitution of Tyr with His at position B16 (B16H) showed a small stabilising effect on human insulin, the substitution of Tyr with Glu at position A14 (A14E) resulted in a substantial increase in stability (Fig. 1d, structures of insulin analogs described in the article are shown in Supplementary Fig. 1 and Supplementary Table 1). A combination of A14E with B25H and B16H (the backbone of insulin icodec) led to a further improvement in stability, while modification of the backbone with C20 fatty diacid and linker had only a small effect on the stability of insulin icodec.

### Thermodynamic stability

The folding stability of the insulin analogues was examined via monitoring of the far-UV circular dichroism (CD) signal of unfolding induced by guanidine hydrochloride (GuHCl). From curve fitting, the midpoint of the unfolding curve was determined and used as an indicator of the thermodynamic folding stability of the monomeric form of insulin analogues[10].

The midpoint of the unfolding curve for human insulin was found at 4.50 M GuHCl (curve shape and midpoint were similar to values reported previously[10]). A substantial right shift of the unfolding curves, a strong indication of folding stabilisation, is observed for analogues with A14E; A14E, B25H; and A14E, B16H, B25H substitutions successively shifting the midpoint to 4.91, 5.01 and 5.10 M, respectively). The stabilising effect of B25H has

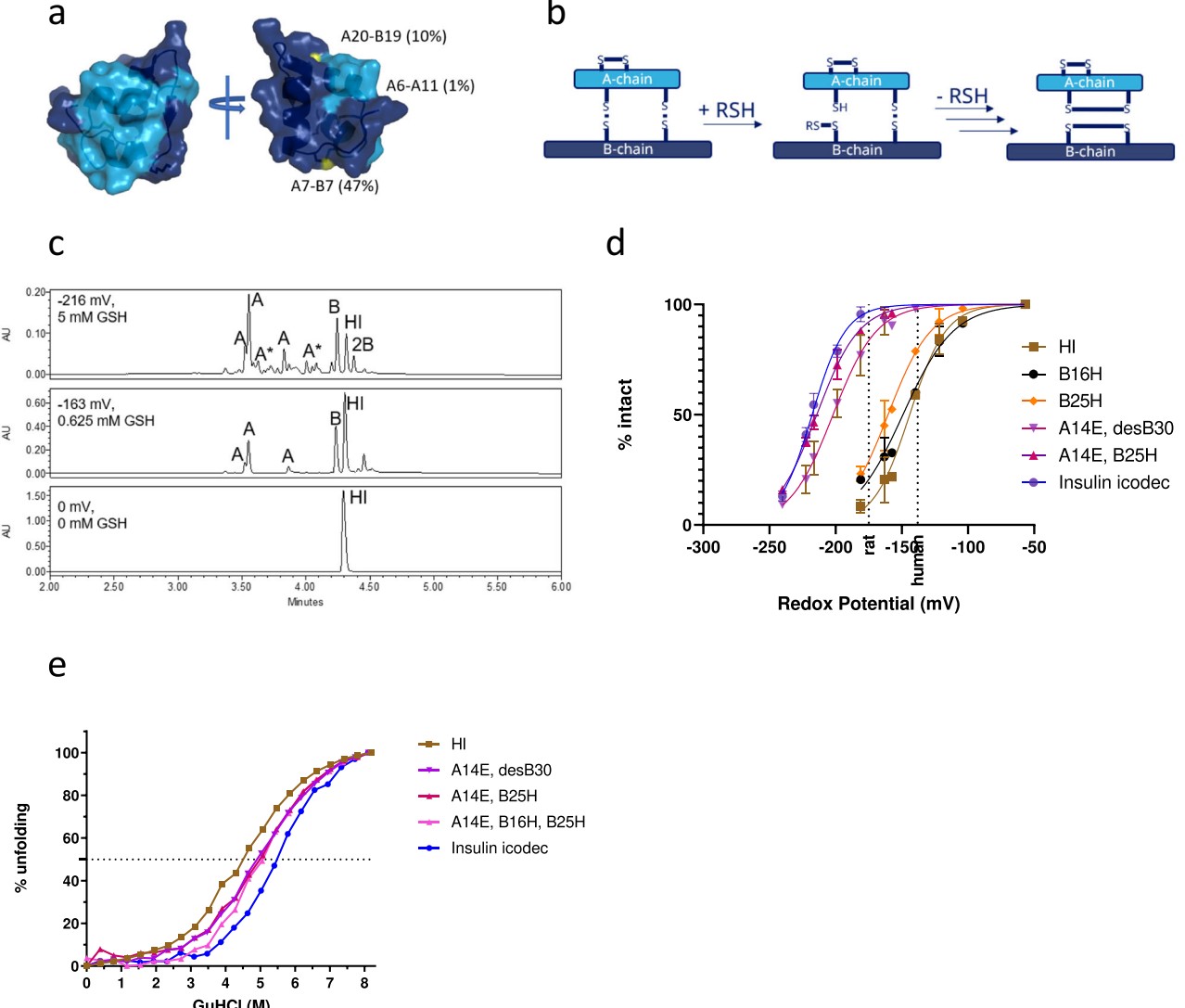

**Fig. 1 | Thiol–disulphide reaction and stability of insulin icodec. a** Solvent exposure of disulphide bonds in insulin icodec. Colour scheme: light blue, insulin A-chain; dark blue, insulin B-chain; yellow, sulphur atoms forming disulfide bonds. Relative solvent exposure for sulphur atoms in each disulphide bond is indicated. **b** Schematic representation of thiol–disulphide exchange reaction leading to insulin chain-splitting. RSH represents a thiol group reacting with a disulphide bond in insulin. **c** High-performance liquid chromatography chromatograms showing human insulin after 4-h incubations at 37 °C under different redox conditions as described in "Methods". Top panel represents the incubation of human insulin without added glutathione, the middle panel shows human insulin incubated with 0.625 mM GSH and 1 mM GSSG representing a state where ~50% of HI is degraded and the bottom panel shows human insulin incubated with 6.3 mM GSH and 1 mM GSSG. A corresponds to A-chain isoforms containing two internal disulfide bonds; B corresponds to B-chain containing an internal disulfide bond; 2B corresponds to two insulin B-chains forming a dimer via two disulfide bonds; HI corresponds to human insulin. Additional peak assignment and MS spectra of the corresponding species are shown in Supplementary Fig. 2. **d** Stability of selected insulin analogues exposed to varying redox potential as described in the Methods section, showing mean ± SD, $n = 3$. See the text for a definition of insulin substitutions. The two dashed lines represent a glutathione redox potential in rat and human plasma, respectively. **e** Stability of selected insulin analogues exposed to varying concentrations of guanidinium hydrochloride as described in "Methods", $n = 1$. Source data are provided as a Source Data file.

previously been reported[10]. The addition of the fatty acid moiety (gGlu-2OEG C20) shifts the unfolding curve to 5.42 M. A similar effect of a fatty acid substitution at B29 was observed for insulin detemir[11]. It has been suggested that this stabilisation of the monomeric analogue is related to the finding that the fatty acid is interacting with the dimer interface of the B-chain[11].

Thus, the observed stabilising effect of insulin icodec substitutions on the disulphide bonds correlated well with the contributions of these substitutions to folding stability (Fig. 1e). These substitutions may reduce solvent exposure of the disulphide bonds of insulin, limiting the chain-splitting reaction by reducing structural flexibility and resulting in the observed correlation.

## Crystal structure of insulin icodec

Next, we determined insulin icodec's crystal structure to find out if the stabilising effects of insulin icodec substitutions have a structural explanation. The crystal structure of insulin icodec contains three icodec molecules in T conformation in the asymmetric unit: two that form an insulin dimer (molecules 1 and 2) and one additional insulin molecule 3 (Fig. 2a and Table 1). When compared to human insulin (Protein Data Bank [PDB] entry 6S34), the conformation of the dimer is basically identical, resulting in a root mean square (RMS) distance of 0.4 Å for equivalent Cα atoms (Fig. 2b). Importantly, side chains for the mutated amino acid residues in positions A14, B16 and B25 occupy roughly the same space as the corresponding side chains in human

**Fig. 2 | X-ray structure of insulin icodec. a** Cartoon representation of the trimeric arrangement of insulin icodec in the crystal. The dimer is coloured in orange and dark blue, for A-chain and B-chain, respectively. The additional icodec molecule is coloured in yellow (A-chain) and purple (B-chain). **b** Superposition of icodec molecule 1 with molecule 3 of the icodec trimer (left) and icodec molecule 1 with human insulin (right). Colours for icodec are as in (**a**); human insulin is in light yellow (A-chain) and cyan (B-chain). Residues A14, B16 and B25, which differ between human insulin and icodec are depicted in stick representation. **c** Interactions (dashed lines) around the A20-B19 disulphide bond for insulin icodec molecule 1 (1st from the left), insulin icodec molecule 3 (2nd from the left), human insulin (3rd from the left) and OI338 (4th from the left).

insulin. Lys B29 is pointing into the solvent, and the fatty acid attached to it is disordered and not visible in the electron density, as has been observed previously[12–14].

Structurally, icodec molecule 3 differs from the human insulin structure in the conformation of both termini of the B-chain. The N-terminus is more helical in molecule 3 so that the central helix starts with Cys7 in molecule 3 instead of Ser9 in molecule 1, which is also what is typically observed in T conformation. The C-terminus of the B-chain deviates from the dimer-forming β-sheet conformation from residue B25H onwards (Fig. 2b). The observed conformational differences between molecule 3 and molecules 1 and 2 are a result of the trimeric arrangement in the crystal.

When looking at the 3D structure around the disulphide bonds, the A20-B19 environment is affected by the substitutions in the icodec molecule. The side chain of Arg B22 is located directly above the A20-B19 disulphide bond and alteration of its conformation and dynamics will change the solvent accessibility of this disulphide bond. In human insulin, this positively charged side chain is interacting via long-range electrostatic interactions with the carboxy group of the A-chain C-terminus and with Glu A17 (Fig. 2c, right).

In insulin icodec, B25H is forming a hydrogen bond with the side chain of Asn A21. This leads to a stabilisation of the electrostatic interaction between Arg B22 and the C-terminal carboxy group of the A-chain (Fig. 2c, first and second panels). On the opposite side of the

molecule, the substitution of Tyr A14 to Glu leads to a stacking of negative charges along the second A-chain helix, effectively resulting in a stronger electrostatic interaction with Arg B22 (Fig. 2c). Insulin icodec was crystallised at pH 4.6 which results in the Arg22 guanidinium having a longer distance to Glu A17 compared to human insulin (Fig. 2c, third panel), indicating that Glu A14 and Glu A17 are partially protonated. In the structure of OI338 (PDB entry 6s4i), which has the same backbone mutations and was crystallised at pH 8.5 the Arg22 to Glu A17 is 2.8 Å (Fig. 2c right panel).

Temperature factors, as determined in X-ray crystal structure analyses, are related to the mean square displacement of an atom and can be used as a measure of atomic flexibility in the crystal. A lower flexibility of the beta turn containing Arg B22 can be seen when analysing normalised temperature factors; the relative average temperature factors of Arg B22 and adjacent residues are lower than those of human insulin, Table 2. We interpret this as Arg B22 having lower flexibility in insulin analogues containing the B25H and A14E mutations. With the amino acid segment containing B22 being more rigid, the A19-B20 disulphide bond is more shielded from the solvent and potentially better protected from reacting with thiol compounds in solvent. The observed lowering in temperature factor is also present in insulin analogue OI338 (PDB entry 6s4i)[12,13] which carries the A14E and B25H backbone substitutions but crystallizes in the same crystal form as human insulin (PDB entry 6s34).

## In vitro plasma stability

With the clear effect observed on insulin stability of the individual substitutions in the in vitro disulphide bond stability assay, we subsequently addressed the effect of the substitutions in a more biological

**Table 1 | Crystal structure of insulin icodec: data collection and refinement statistics**

| | |
|---|---|
| **Wavelength (Å)** | 1.5418 |
| **Resolution range (Å)** | 29.67–2.0 (2.07–2.0) |
| **Space group** | P 4₃ 21 2 |
| **Unit cell (Å, deg)** | 54.793 54.793 138.355 90 90 90 |
| **Total reflections** | 27,175 (2730) |
| **Unique reflections** | 14,970 (1461) |
| **Multiplicity** | 1.8 (1.9) |
| **Completeness (%)** | 99.57 (98.91) |
| **Mean I/sigma (I)** | 28.48 (3.43) |
| **Wilson B-factor (Å²)** | 35.35 |
| **R-merge** | 0.0206 (0.1751) |
| **R-meas** | 0.0291 (0.2476) |
| **R-pim** | 0.0206 (0.1751) |
| **CC$_{1/2}$** | 0.999 (0.972) |
| **CC\*** | 1 (0.993) |
| **Reflections used in refinement** | 14,918 (1449) |
| **Reflections used for R-free** | 1491 (145) |
| **R-work** | 0.178 (0.224) |
| **R-free** | 0.217 (0.249) |
| **CC (work)** | 0.952 (0.949) |
| **CC (free)** | 0.917 (0.915) |
| **Number of non-hydrogen atoms** | 1217 |
| Macromolecules | 1159 |
| Ligands | 14 |
| solvent | 50 |
| **Protein residues** | 147 |
| **RMS (bonds) (Å)** | 0.011 |
| **RMS (angles)** | 1.05 |
| **Ramachandran favoured (%)** | 99.26 |
| **Ramachandran allowed (%)** | 0.74 |
| **Ramachandran outliers (%)** | 0.00 |
| **Rotamer outliers (%)** | 0.74 |
| **Clashscore** | 4.88 |
| **Average B-factor (Å²)** | 47.71 |
| Macromolecules | 47.57 |
| Ligands | 56.44 |
| Solvent | 49.61 |
| **PDB accession code** | 8RRP |

setting and carried out a series of in vitro plasma stability experiments in rat plasma. We found that the individual B16H, B25H and A14E substitutions have a stabilising effect on the insulin backbone, with the A14E substitution having the most pronounced effect of the three (Fig. 3a). The results correlated well with those from the in vitro disulphide bond stability assay (Fig. 1d). The stabilising effects of the substitutions is apparently additive in nature, and insulin analogues containing either the two A14E and B25H substitutions (pink triangles, C18 fatty diacid, and red squares, C20 fatty diacids, in Fig. 3a) or all three substitutions (blue circles in Fig. 3a), resulting in insulin icodec being stable in this experiment. The appearance of insulin B-chain containing the intramolecular disulphide bond together with a lack of other B-chain degradation products provides unequivocal support for the hypothesis that the thiol–disulphide exchange reaction is responsible for the disappearance of intact insulin (Fig. 3b). The detection of the free B-chain, and the fact that we were unable to identify any proteolytic degradation products in these incubations, is in contrast to incubating human insulin in rat plasma, which rapidly disappears and only proteolytic degradation products and no free B-chain being detected (Supplementary Fig. 4).

To investigate potential interspecies differences, we also performed in vitro incubations in plasma from dogs, minipigs or humans (Fig. 4a–c). As seen from the results, the overall trends and effects of the substitutions are also observed in plasma of these other species. The stability of the insulin molecules increased in the following order: rat plasma, minipig and human plasma, and dog plasma (most stable). We hypothesise that this could be a result of distinct redox potentials of the specific plasma types in this static experimental set-up, making insulin molecules either more or less stable in the specific plasma types. Our results obtained in the different plasma types (Figs. 3 and 4) indicate that the undesired degradation via insulin chain-splitting is highest in rats, followed by in minipigs, humans and dogs. Therefore, incubations in rat plasma offer the highest resolution for measuring the differences in disulphide bond stability when introducing substitutions in the insulin backbone. No insulin icodec metabolites originating from proteolytic degradation were detected in the plasma samples, suggesting that the degradation pathway of insulin icodec in plasma incubations is predominantly via chain-splitting.

## In vivo minipig pharmacokinetics

Next, we sought to demonstrate the relevance of this non-receptor-mediated clearance mechanism in the dynamic setting of an in vivo minipig study. The exposure profile of insulin icodec following intravenous dosing of minipigs shows a terminal half-life of about 50 h (Fig. 5, circles). In addition to intact insulin icodec as the main component, we were also able to identify and quantify free insulin icodec B-chain (Fig. 5, squares). The plasma half-life of the B-chain is unknown; however, considering the lack of receptor-mediated clearance and limited renal clearance of the B-chain due to albumin association, it would be expected to be longer than that of insulin icodec. In contrast, we were not able to detect free insulin icodec A-chain in the plasma samples, presumably because of fast disappearance of the A-chain either via renal

**Table 2 | Temperature factors in X-ray crystal structure**

| | Average original temperature factor for B-helix (B9–19) used for normalisation [Å²] | Average normalised temperature factor for residues 20–23 (all atoms) [Å²] | Average normalised temperature factor for residues 20–23 (peptide backbone atoms) [Å²] |
|---|---|---|---|
| Human insulin 6S34 | 19.2 | 35.0 | 24.3 |
| Icodec Mol1 | 37.0 | 25.8 | 18.7 |
| Icodec Mol2 | 39.6 | 29.8 | 20.0 |
| Icodec Mol3 | 60.5 | 21.3 | 17.0 |
| Insulin OI338 (PDB 6S4I) | 29.6 | 26.3 | 21.2 |

clearance due to the negligible affinity to albumin and/or via proteolytic degradation. Furthermore, the mass spectrometry (MS) ionisation efficiency and resulting detection limit is poor for the A-chain owing to the lack of basic amino acid residues. No insulin icodec metabolites originating from proteolytic degradation were detected in the samples, suggesting that the degradation pathway of circulating insulin icodec in minipigs is predominantly via chain-splitting.

## Insulin icodec metabolism in humans

In the final experiment, we analysed serum samples from patients with type 2 diabetes who received five weekly doses of insulin icodec. Insulin icodec was the principal serum component, and the serum exposure of the free B-chain (with internal disulphide bond between B7 and B19) was ~10% of that of intact insulin icodec (Fig. 6a). Two additional minor metabolites were identified as products from proteolytic degradation of

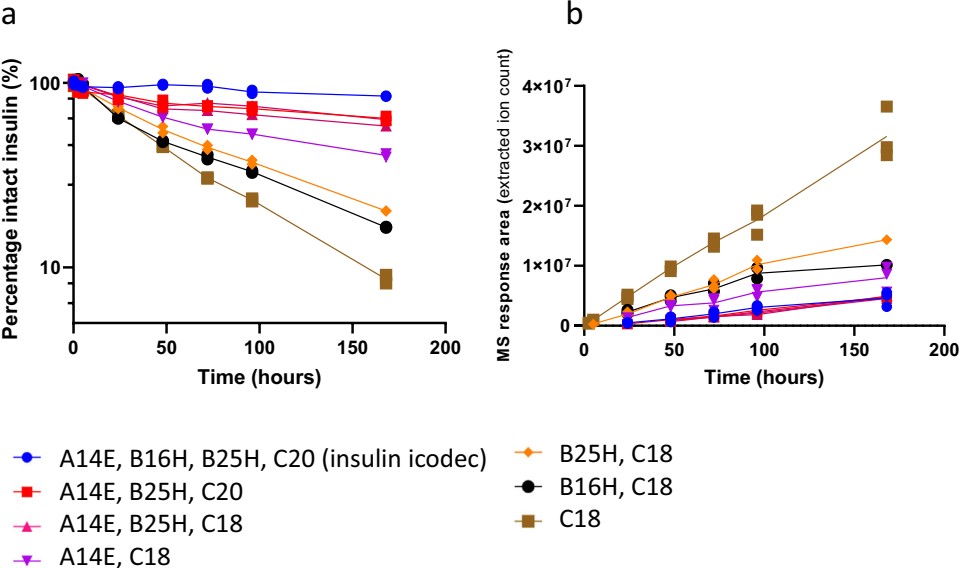

**Fig. 3 | In vitro rat plasma stability of selected insulin analogues.** The disappearance of intact insulin analogues upon incubation in rat plasma (**a**), with the corresponding appearance of insulin B-chains (**b**), *n* = 2. All insulin B-chains contain an internal disulphide bond connecting Cys 7 with Cys 19, assessed by the exact monoisotopic masses. See text for definition of insulin substitutions. Source data are provided as a Source Data file.

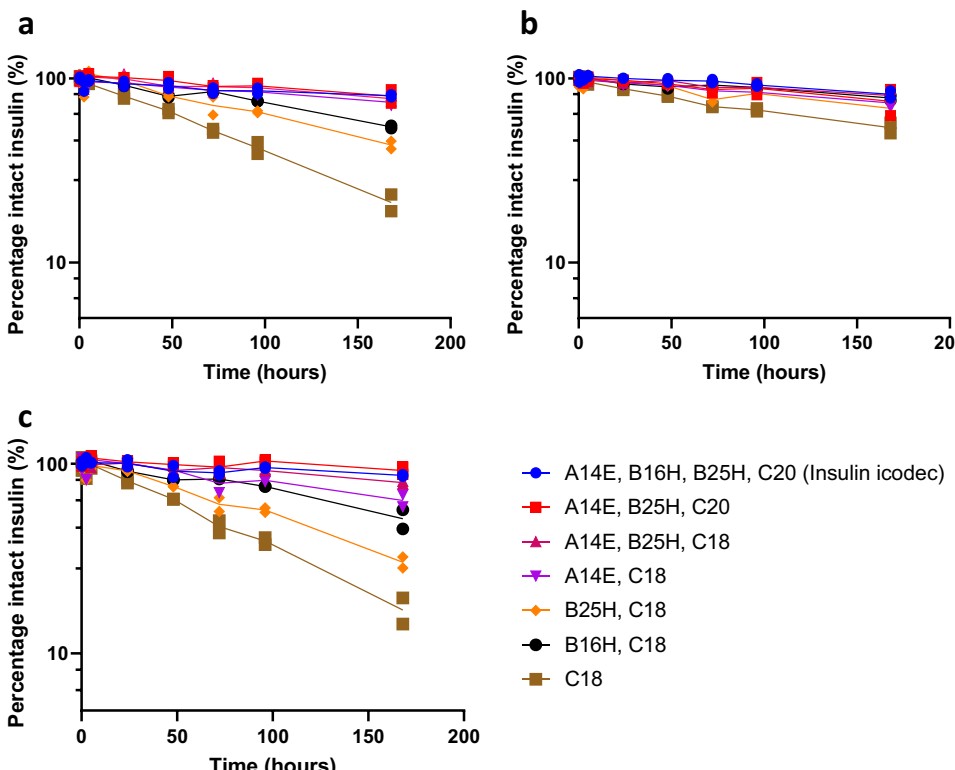

**Fig. 4 | In vitro plasma stability of selected insulin analogues.** Disappearance of intact insulin analogues upon incubation in human (**a**), dog (**b**) and minipig (**c**) plasma, *n* = 2. The corresponding appearance of B-chains are shown in Supplementary Fig. 5. See text for definition of insulin substitutions.

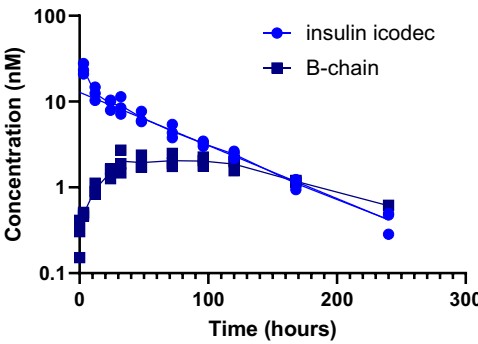

**Fig. 5 | Pharmacokinetics of insulin icodec administered intravenously to minipigs.** Disappearance of insulin icodec (circles) and appearance of insulin icodec B-chain (containing a disulphide bond connecting Cys 7 and Cys 19; squares) in plasma are shown, $n = 3$. Source data are provided as a Source Data file.

the B-chain (Fig. 6b): B29 and B24–29, with serum exposure of ~3% and less than 1% of that of intact insulin icodec, respectively. It is unclear whether these degradation products are formed directly from insulin icodec or are further degradation products of the insulin icodec B-chain generated by chain-splitting. As expected, the area under the concentration–time curve of the free B-chain and metabolites (all with intact fatty acid side chain) was significant in serum owing to continuous formation from circulating insulin icodec, the lack of receptor-mediated clearance and renal clearance that was hampered by its fatty acid-mediated albumin-binding.

## Discussion

Compared to a group of insulin analogues with similar insulin receptor affinity, insulin icodec demonstrated superior plasma half-life and glucodynamic potency when tested in a preclinical rat model[4]. We hypothesised that improved plasma stability was the explanation for the beneficial properties of insulin icodec[4]. Improvements in plasma stability must be mediated by the structure of insulin icodec, and thus the three amino acid substitutions (Phe with His at position B25 [B25H], Tyr with His at position B16 [B16H] and Tyr with Glu at position A14 [A14E]) and/or the fatty diacid modification (C20 diacid including the 2x OEG and gGlu linker) provide such molecular stability. These amino acid substitutions were discovered in our earlier efforts to optimise insulin for oral absorption and were shown to reduce insulin susceptibility for degradation by gut enzymes[12,13]. However, since we have never observed relevant levels of metabolites resulting from proteolytic cleavage of fatty acid-protracted insulin analogues in plasma after subcutaneous dosing, we focused on metabolites resulting from the degradation of disulphide bonds.

Solvent-exposed disulphide bonds readily undergo reduction and/or thiol– disulphide exchange at neutral or alkaline pH in the presence of free thiol groups. For example, hirudin[15] and ribonuclease A[16] can be fully reduced with a 0.5 mM solution of dithiothreitol (DTT) at 23 °C within 20 min. In contrast, reduction of the interchain disulphide bond keeping together the A-chain and B-chain of α-thrombin requires a 100-fold higher DTT concentration. This demonstrates that the surrounding protein structure can significantly alter the reactivity of disulphide bonds in proteins[17–20].

Here, we describe a set of in vitro experiments showing that amino acid substitutions in the insulin sequence have profound effects on the susceptibility of insulin for the thiol–disulphide exchange reaction. The A14E substitution had the most powerful stabilising effect, while B25H and B16H substitutions had only a modest effect on disulphide bond stability in insulin. Furthermore, we have shown that the disulphide bond stabilising effect of these substitutions correlates well with the propensity to prevent unfolding of the insulin molecule in the presence of GuHCl. This correlation suggests that improving the

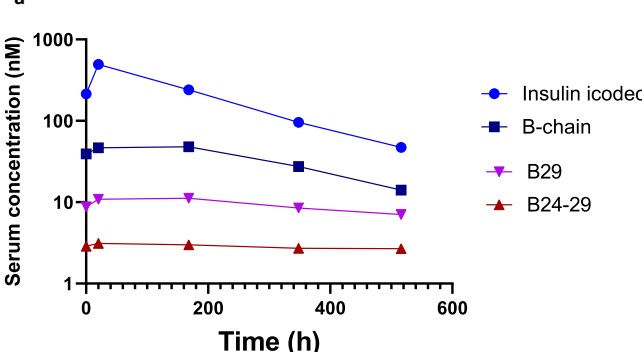

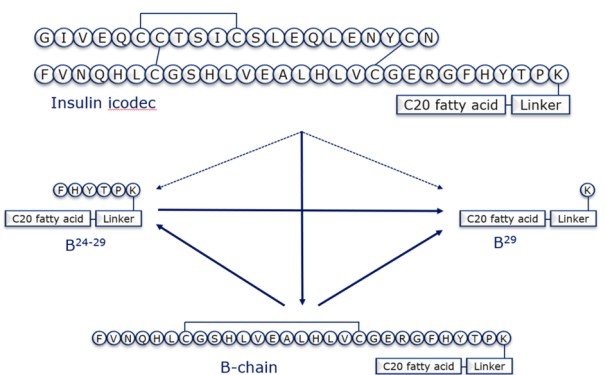

**Fig. 6 | Pharmacokinetics of insulin icodec and major metabolites in human serum samples. a** Pharmacokinetic profiles of insulin icodec (light blue circles), insulin icodec B-chain (containing a disulphide bond connecting Cys 7 and Cys 19; dark blue squares), B29 acylated metabolite (purple inverted triangles) and B24–B29 acylated metabolite (brown triangles) at steady state (after the fifth weekly subcutaneous dose) in people with diabetes. Each time point represents a single value obtained from a pooled plasma sample from 12 human subjects.
**b** Possible metabolic pathways leading to formation of the observed metabolites (Supplementary Table 3). The solid lines represent the most likely paths while the dashed lines represent alternative paths to the observed degradation products of insulin icodec. Source data are provided as a Source Data file and in Supplementary Table 4.

folding stability of insulin protects disulphide bonds in this molecule from solvent exposure and, consequently, prevents thiol–disulphide exchange reaction with thiol compounds in blood. Decreased flexibility of insulin icodec was also supported by the analysis of the X-ray structure, specifically showing an increased interaction network between the C-termini of the A-chain and B-chain. It is interesting to note that all insulin molecules tested in clinical development for once-weekly therapy contain substitutions of the solvent-exposed tyrosine residue at A14 with acidic amino acid residues: A14D in insulin efsitora alfa[21] and A14E in ^LAPS^Insulin115[22].

Our observations of insulin degradation in plasma by the thiol–disulphide exchange reaction are not surprising because blood contains greater than 0.5 mM of free thiol groups originating from, for example, glutathione, cysteine and plasma proteins[23] and maintains a slightly alkaline pH, satisfying the necessary conditions for this reaction to proceed. Indeed, incubation of bovine insulin in human plasma was previously shown to result in slow chain separation[24]. Similarly, changes in the disulphide bond pattern of therapeutic antibodies, probably caused by thiol–disulphide exchange reaction, were also observed in plasma[25], and disulphide bond scrambling was shown to take place for conotoxin analogues in both rat plasma and human plasma despite these analogues being stabilised towards proteolytic degradation[26,27]. The appearance of insulin chains containing

disulphide bonds during plasma incubations is indeed indicative of the thiol–disulphide scrambling-based degradation mechanism.

Surprisingly, the overall rates of insulin disappearance after incubation in plasma from different species were different. The fastest disappearance rate was observed in rat plasma followed by in minipig plasma and human plasma, while the disappearance rate was slowest in dog plasma. The A14E substitution alone was sufficient to stabilise the insulin molecule to a comparable level to that for insulin icodec in dog or human plasma. However, insulin icodec was more stable than insulin containing only the A14E substitution in rat or minipig plasma. This suggests that B16H and B25H substitutions are relatively more important to stabilise insulin towards thiol–disulphide exchange-based degradation in rat or minipig plasma than in dog or human plasma. One explanation for these observations could be that there are interspecies differences in plasma thiol content. To our knowledge, comprehensive analysis of reduced and oxidised thiol species is only available in human plasma[23]. Reduced and oxidised glutathione species were also measured in rat plasma[28], but no information regarding other thiol species was included. Nonetheless, the redox potential of glutathione in rat plasma (−175 mV) is significantly lower than that in human plasma (−137 mV). This difference in glutathione redox potential at least partially explains the relatively lower stability of insulin analogues in rat plasma in comparison to in human plasma. Furthermore, different thiol redox couples (including redox-active enzymes) could also contribute to an explanation of the observed species differences.

The results from the plasma incubations suggest the importance of the thiol-based degradation pathway for long-acting insulin analogues. Given that this pathway degrades insulin molecules before they elicit their effect via the insulin receptor, any such degradation will reduce both the circulating half-life and glucodynamic potency of insulin. The observed plasma degradation rates are relatively slow, with half-lives ranging from ~1 day to several weeks (Figs. 3 and 4). It is likely that the in vivo potency of insulin molecules in general will depend on both the disulfide bond stability of the individual molecule and its circulating half-life, and this pathway could have a significant effect on insulin analogues with ultra-long circulating half-lives.

Both our in vitro assay and in vitro plasma incubations could be regarded as steady-state assays because the composition of the system is constant during the incubation, with the exception of the negligible effects caused by air on top of the solution in the vials/plates. The steady-state conditions are also supported by the linear disappearance of intact insulin molecules throughout the week-long incubations in plasma (Figs. 3 and 4). However, conditions are different in living organisms, in which insulin concentrations are affected by receptor-mediated clearance, renal clearance, distribution between blood and tissues and a constant flux of reduced and oxidised compounds from the cells to blood and back. Nonetheless, our results show that insulin is also degraded by thiol–disulphide exchange reaction in a living organism. It is difficult to estimate the actual degradation rate, because detailed understanding of the pharmacokinetic parameters of all the degradation products is not available. However, thiol–disulphide exchange degradation and proteolytic degradation rates for insulin icodec in human plasma can be compared from the observed levels of the three inactive metabolites found in human plasma (Fig. 6a). None of these can be cleared by insulin receptors, and all of them contain the same albumin-binding fatty acid moiety; therefore, in humans, the rate of plasma thiol–disulphide-based degradation of insulin icodec as estimated by the appearance of the insulin icodec B-chain is at least fourfold faster than the proteolytic degradation (acylated B24–29 and B29, see Fig. 6). The actual rate difference could be much higher if the proteolytic degradation products of the B-chain result predominantly from further degradation of the B-chain already formed by the thiol–disulphide exchange reaction.

Our results clearly demonstrate that insulin in rat, minipig, dog or human plasma is degraded by thiol–disulphide exchange reaction.

Therefore, any disulphide bond-containing compound present in plasma will potentially be subject to this degradation pathway. Insulin degradation via this pathway is unique, because the end-products (insulin A-chain and B-chain) are easy to distinguish from intact insulin in plasma by the analytical methods used for pharmacokinetic analysis. To assess whether the thiol–disulphide exchange reaction is involved in plasma degradation of a compound requires that the detection methods used must be able to distinguish native disulphide bonds from non-native ones, a non-trivial task requiring high-resolution analytical techniques and understanding of which forms of the molecules are detected with a given method. It is interesting to note that many protein therapeutics, such as monoclonal antibodies, coagulation factors, growth hormone etc., contain several disulphide bonds and could in principle be degraded by this pathway.

In summary, we provide an unequivocal body of evidence that thiol–disulphide exchange reaction-based degradation of insulin is responsible for the largest degradation products of insulin in plasma (Fig. 7). Our results also demonstrate that the substitutions reducing the rate of this degradation significantly contribute to the long pharmacodynamic effects of insulin icodec. Therefore, multiple parameters need to be optimised and balanced in designing efficacious once-weekly insulin molecules with a good safety profile; not only, as described previously, by modifying insulin receptor affinity and albumin-binding affinity but also limiting non-receptor-mediated clearance to retain a clinically relevant pharmacodynamic potency.

## Methods

### Ethics statement
All people living with diabetes participating in the clinical trial signed a written consent form. Trial protocol, amendments, subject information and consent forms were reviewed and approved by an independent ethical committee Ärztekammer Nordrhein, Körperschaft des öffentlichen Rechts, Ethikkommission, Germany. Animal experiments were performed according to permission granted by the Animal Experiments Inspectorate, Ministry of Environment and Food of Denmark.

### Production of insulin icodec
All insulin analogues were expressed in *Saccharomyces cerevisiae*, purified and modified by attaching fatty acids to insulin B29K epsilon amino group as described previously[4].

### Crystal structure determination of insulin icodec
An aqueous solution of insulin icodec at 39 mg/mL and 20 mM magnesium acetate was equilibrated against a reservoir containing 0.02 M calcium chloride, 0.1 M sodium acetate trihydrate (pH 4.6) and 30% (v/v) (+/−)-2-methyl-2,4-pentanediol. Tetragonal bipyramidal crystals grew within a few days.

The crystals were flash-frozen after cryoprotection in a solution containing the reservoir solution plus 20% (v/v) ethylene glycol. Diffraction data were collected on a Rigaku MicroMax-007 HF rotating anode X-ray generator equipped with a MarResearch 165 charge-coupled device camera. XDS[29] was used for data reduction and scaling. The structure of insulin icodec was solved by molecular replacement using PDB entry 1B2F as search model. The crystal contains three molecules per asymmetric unit.

Iterative refinement and rebuilding with Coot[30] yielded the final model. All calculations were carried out with the Phenix suite of programs[31]. The final model was deposited with the PDB under accession code 8RRP. A stereo view of the representative part of the electron density is shown in Supplementary Fig. 6.

### In vitro stability as a function of redox potential
Insulin analogues (10 μM) were incubated in 100 mM phosphate buffer (pH 8.0) with different redox potentials for 4 h at 37 °C. Redox

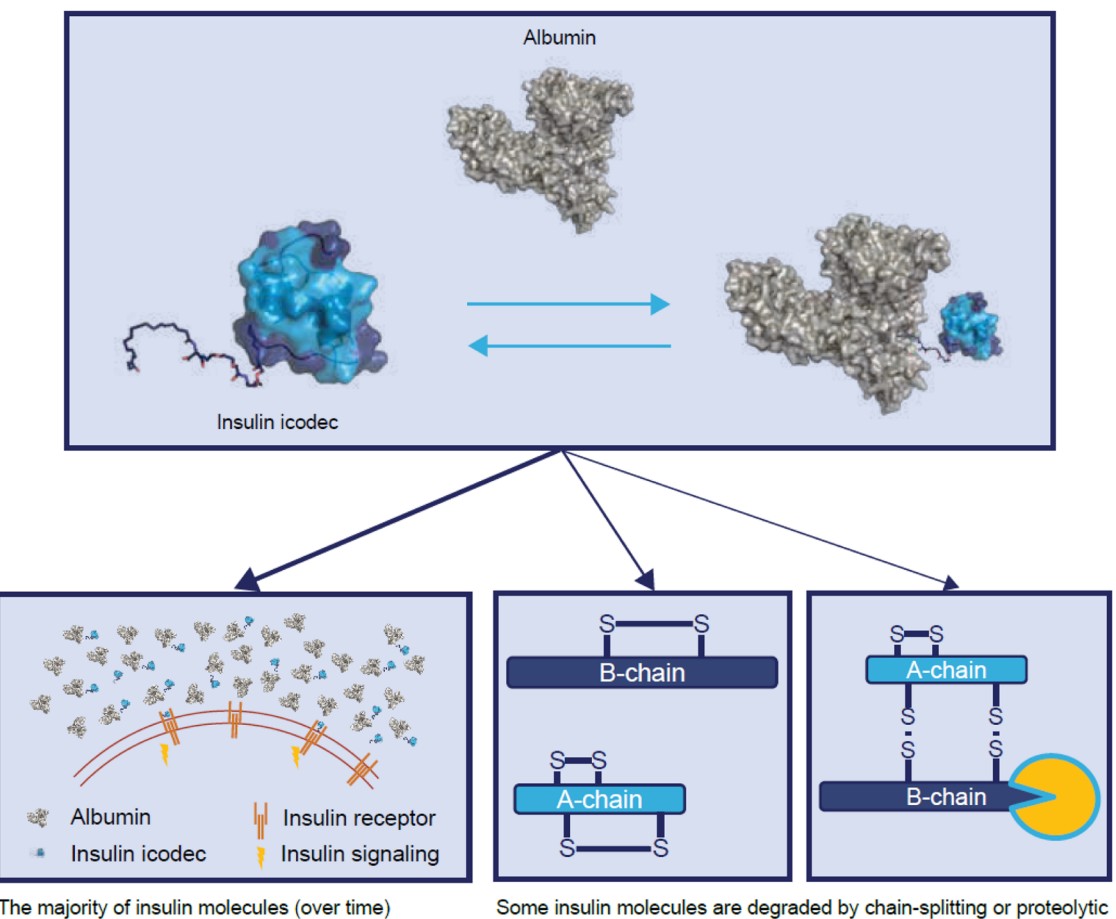

**Fig. 7 | Fate of insulin icodec in plasma.** Scheme showing the fate of insulin icodec in plasma.

potential was varied by increasing the concentration of reduced glutathione (0–25 mM) in the presence of a fixed concentration of oxidised glutathione (1 mM). Redox potential was calculated using the Nernst equation[32]. The experiments were performed using the OT-2 liquid handler (Opentrons, Brooklyn, NY, USA). Remaining intact insulin was quantified using reversed-phase ultra-high-performance liquid chromatography on a BEH C4 column (1 ×50 mm, Waters Corp, Milford, MA, USA) with a linear gradient of acetonitrile in 0.1% trifluoro acetic acid (TFA). HPLC chromatograms and MS spectra of observed degradation products are shown in Supplementary Fig. 2 and Supplementary Table 2. Results were normalised to 0 mM glutathione, plotted as a function of redox potential using GraphPad Prism (version 9.01) and fitted with a non-linear regression model (Absolute IC50) with the following constrains (Top = 100, Baseline = 0 and Bottom = 0). Human insulin was used as a standard in each assay.

**Thermodynamic stability**
Thermodynamic stability of insulin analogues was measured by following change in the CD spectra in the far-UV region of insulin analogues after incubation with increasing concentrations of GuHCl.

Stock solutions of insulin analogues were prepared in 10 mM Tris/perchloric acid ($HClO_4$) (pH 8.0). Concentrations of the stock solutions were determined using $E276 = 6.20 \times 10^3\,M^{-1}\,cm^{-1}$ for analogues with four tyrosine residues[10]. Under the assumption that all tyrosine residues contribute equally to the absorbance intensity, $E276 = 4.65 \times 10^3\,M^{-1}\,cm^{-1}$ and $E276 = 3.10 \times 10^3\,M^{-1}\,cm^{-1}$ were applied for analogues with three and two tyrosine residues, respectively.

The concentration of GuHCl stock solution in 10 mM Tris/HClO$_4$ (pH 8.0) was determined by refractive index measurement at 25 °C as described[33] using an Abbemat 550 refractometer (Anton Paar, Graz, Austria). GuHCl and Tris base were obtained from Sigma. A pH value of 8.0 was chosen to diminish any effect of insulin dimer dissociation that could create a three-state process and complicate the data-evaluation procedure.

After mixing insulin and GuHCl stock solutions to a resulting concentration of 5 μM insulin and 22 different concentrations of GuHCl in the range 0–8.3 M, samples were equilibrated overnight at 22 °C prior to measurement. CD spectra were obtained using a Chirascan ACD spectrometer (Applied Photophysics, Leatherhead, UK), using a 5 mm path length and scanning the range 214–260 nm with 2 pt/nm at 25 °C. Each sample was scanned three times and the average was reported (Supplementary Fig. 2). Appropriate buffer spectra were subtracted from the CD, and absorbance spectra obtained using Chirascan ACD spectroscopy and subsequently the CD spectra were normalised to the absorbance intensity to adjust for small variations in the concentration of the insulin sample in the flow cell to allow for a global fit procedure. All data points in the range 218–228 nm (21 spectra and 22 different GuHCl concentrations for each analogue) were used for a global fit as described[34].

**In vitro plasma stability and quantification of in vivo samples by liquid chromatography with MS (LC-MS)**
In vitro plasma stability was assessed by incubation of 1 μM insulin analogue in 80% pooled plasma from humans, minipigs, dogs or rats (all mixed male and female from BioIVT, Westbury, NY, USA) and 20% PBS

(pH 7.4) at 37 °C with shaking. At selected time points (0.25, 2.5, 5, 24, 48, 72, 96, 168 h) one volume of the incubation mixtures was subjected to protein precipitation using three volumes of ethanol, followed by centrifugation and dilution of one volume of the supernatant with one volume of water containing 1% formic acid (FA) before LC-MS analysis.

For quantification of the in vivo minipig samples, selected plasma standards with either intact insulin icodec, or the insulin icodec B-chain reference standard with an internal disulfide bond connecting Cys 7 with Cys 19, at concentrations in the range 0.05–200 nM were prepared. The standards were prepared by spiking blank plasma from Göttingen Minipigs. Prior to LC-MS analysis, the plasma samples (blank plasma, standards and study samples) were prepared by plasma protein precipitation. Protein precipitations were conducted by adding three volumes of ethanol to one volume of plasma. The samples were centrifuged, and one volume of supernatant was mixed with three volumes of water containing 1% FA.

The LC-MS analysis was carried out using a TurboFlow high-performance liquid chromatography system from Thermo Fisher Scientific (Bremen, Germany) coupled to a Q Exactive Orbitrap mass spectrometer. The liquid chromatography mobile phases consisted of solvent A: Milli-Q water with 5% organic solvent (50% methanol/50% acetonitrile) and 1% FA; and solvent B: Milli-Q water with 95% organic solvent (50% methanol/50% acetonitrile) and 1% FA. For quantification of the in vivo samples a TurboFlow Cyclone $0.5 \times 100$ mm column from Thermo Fischer Scientific (Bremen, Germany) was used for extraction, before analytical elution on an XSelect Peptide CSH C18, 3.5 μm, $2.1 \times 50$ mm column (controlled at 60 °C) from Waters (Wilmslow, UK) using a flow rate of 400 μL/min and a linear 40% gradient of mobile phase solvent B from 50% to 90% over 2.5 min. The samples from the in vitro experiments were directly loaded onto the analytical column (XBridge Protein BEH C4, 3.5 μm, $2.1 \times 50$ mm, 300 Å from Waters, Wilmslow, UK) and eluted using a linear gradient of 10–90% B over 5 min. The Orbitrap mass spectrometer was operating in positive ionisation mode with a spray voltage of 3.7 kV, and a resolution of 35 K using $m/z$ 500–2000 full scan mode for the in vitro samples and single ion monitoring scan mode using 5 $m/z$ isolation windows of the most abundant charge state of the peptides from the in vivo samples.

The LC-MS data were processed and quantified using the Quan Browser in the Xcalibur software from Thermo Fisher Scientific (Bremen, Germany) with a lower limit of quantification of 0.2 nM for the in vivo samples.

Plotting of data and calculation of half-life was done in Prism (version 9.0.1, GraphPad Software, Boston, MA. USA).

### In vivo minipig pharmacokinetics

Three female Göttingen Minipigs (Ellegaard Göttingen Minipigs, Dalmose, Denmark) weighing ~22 kg were included in the study. Animals were fed chow once daily. Minipigs were individually housed with the possibility for snout-to-snout contact with neighbouring minipigs. The minipigs always had bedding and straw as enrichment and free access to water. Room temperature was maintained at $22 \pm 2$ °C, relative humidity was in the range 30–70% and artificial light was provided from 6:00 AM to 6:00 PM. Windows allowed natural light in addition to the mentioned light cycle. Animals were instrumented with central venous catheters for blood sampling. Insulin icodec was dosed intravenously (2 nmol/kg) and blood was sampled frequently during the first 36 h and then daily until 10 days after dosing for the measurement of plasma exposure.

### Identification and quantification of human serum metabolites

Serum from 12 men living with type 2 diabetes who received multiple subcutaneous doses of 24 nmol/kg/week of insulin icodec (Novo Nordisk Trial ID: NN1436-4314, ClinicalTrials.gov ID: NCT02964104) collected at 0, 20, 168, 348 and 516 h after the fifth weekly dosing were selected for LC-MS analysis. Equal volumes of serum samples from each selected time point were pooled. One volume of each serum pool was mixed with 2 volumes of acetonitrile:methanol 4:1 (v/v) during vortex mixing followed by centrifugation at $5000 \times g$ for 5 min to spin down the precipitate. After centrifugation, 1 volume of the supernatant was transferred to a vial and diluted with 1 volume of water and analysed by LC-MS consisting of an Acquity I-Class ultra-performance liquid chromatography (UPLC; Waters Corp, Milford, MA, USA) and MS (Synapt G2-S time-of-flight mass spectrometer, Waters Corp, Milford, MA, USA) for explorative analysis of potential metabolites from insulin icodec. After analysis and deconvolution of the MS data, to generate their molecular masses, the data were matched to data from in silico structures of potential metabolites from insulin icodec and metabolite references. Metabolites with mass accuracies of less than 10 ppm compared with reference masses were considered as acceptable for confirming structure identity. The identified serum metabolites were then quantified using UPLC and tandem MS in an analysis of all serum pools and with metabolites references as calibration standards (Supplementary Tables 3 and 4). The pharmacokinetic parameters of the serum components (insulin icodec and metabolites) were then calculated by non-compartmental analysis for evaluation of exposure of the individual components from area under the curves (AUCs), with $AUC_{0-168h}$ as the exposure over a dosing interval and $AUC_{last}$ as the exposure to the last time point (Supplementary Table 5). In addition, the total sum of and the percentages of total AUCs were calculated. Lower limits of quantification for the reported metabolites were 10,000 pmol/L for insulin icodec, 2000 pmol/L for the B-chain, 1000 pmol/L for the B24–29 metabolite and 1000 pmol/L for the B29 metabolite.

### Reporting summary

Further information on research design is available in the Nature Portfolio Reporting Summary linked to this article.

## Data availability

The authors declare that all data supporting the findings of this study are available within the manuscript. X-ray structures with the following access codes were used in this study 6S4I and 6S34 Crystal structure coordinates and structure factors for insulin icodec structure were deposited in the Protein Data Bank (PDB) under the following accession code 8RRP. Source data are provided with this paper.

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

## Acknowledgements

The authors would like to acknowledge the technical assistance provided by Kira Meyhoff-Madsen, Claurice Haugaard, Merete Hvidt, Liu Wei, Susanne Gyldenløve, Anja Benfelt, Carsten Stokkebye Stenvang, Lene Drube, Sebrina Oftedal, Lotte Badeby, Gitte Norup and the animal caretakers. Dr. Janos Tibor Kodra, Dr. Martin Műnzel and Dorte Egholm Jensen are acknowledged for production of some compounds used in this study. This work was funded by Novo Nordisk A/S.

## Author contributions

F.H., C.N.C., E.N. and T.B.K. led and coordinated the studies and drafted the manuscript. F.H. designed the in vitro stability assay. C.N.C. analysed compounds in plasma. D.B.S. evaluated folding stability of insulin analogues. H.H. analysed insulin icodec and its metabolites in human plasma samples. E.J. and G.S. determined the crystal structure of insulin icodec. J.S. designed and performed the minipig experiments. All authors read and contributed to the editing of the paper.

## Competing interests

All authors are present employees of Novo Nordisk A/S and are shareholders of Novo Nordisk A/S.
