## [Peer Review File · Nature Communications]

Reviewers' Comments:

Reviewer #1:

Remarks to the Author:

This is an excellent work reported by the authors about an underappreciated mechanism for insulin clearance in vivo. This is indeed a special circumstance as insulin normally gets cleared way before the disulfide shuffling can happen in blood. I thought this study is clearly presented with convincing evidence to support the conclusion. I am surprised and satisfied to see that the different mutations and fatty acid attachment can lead to different degrees of stabilization in GnHCl condition or redox condition.

I really do not have anything to add and would be happy to see this work published as is.

On the other hand, while I think this may be out of the scope of this work, the authors may want to explore insulin analogs where the disulfide bonds are replaced with non-reducing bonds such as thioether bonds. Will be informative to see if the the existence of insulin B chain is still present.

Reviewer #2:

Remarks to the Author:

This paper reports a new insulin clearance mechanism, i.e., splitting insulin into its A-chain and B-chain by a thiol-disulphide exchange reaction. Insulin icodec, an insulin analogue in clinical development for once-weekly dosing, seems to be protected against this clearance mechanism, although not completely, which contributes to its long half-life.

Identification of this new clearance mechanism may have a significant scientific impact, particularly for the development of insulin analogues with a long half-life.

The paper is well written and the results are convincing.

This reviewer only has minor comments:

1. Page 2, last sentence: The impact of insulin icodec on glucose control (HbA1c) has not been consistent in the phase 3 programme. ONWARDS 6 reports less improvement in HbA1c with icodec vs degludec in people with type 1 diabetes. It might be the best to delete this sentence as the glucose-lowering effect of icodec is of little relevance for the purpose of this paper.

2. Methods: It would be good to learn more about the sensitivity (detection limits) of the LC-MS for the different metabolites.

3. The authors were able to detect free icodec B-chains in the plasma of mini-pigs, whereas free A-chains were not seen. The authors explain this with renal clearance of free A-chains (renal clearance of B-chains is supposed to be limited). It would be great to see some supporting data with A-chain and B-chain measurements in urine. Or are concentrations too low (and the assay too insensitive) to detect the chains in urine.

4. The authors show that "the rate-limiting step of the thiol-disulphide-based reaction is the free thiol (glutathione) attack of a disulphide bond". May this be of clinical importance, e.g. in people taking glutathione supplements?

5. The authors understandably focus on the importance of the thiol based degradation pathway for long-acting insulin analogues, as the degradation rates are too slow for insulins with a short half-life. What about intermediate-acting analogues such as detemir? Can the reduced potency of detemir be partly explained by a higher thiol based degradation pathway compared to other analogues?

Reviewer #3:

Remarks to the Author:

This manuscript describes the proposed identification of a novel mechanism by which insulin is degraded. Discussed are the implications for regulation of circulating insulin and long acting insulin

analogues via this degradation mechanism. It is argued that whilst this mechanism likely only contributes in a minor way to regulation of insulin it has potentially significant impact on circulating long acting analogues (using icodec as the model in this case).

The manuscript provides evidence for the existence of "split" A and B chains generated through thiol-disulphide exchange, leading to formation of new intrachain disulphide bonds in both chains. Such a mechanism for insulin degradation is truly novel and has important implications for future long-acting insulin analogue design. Whilst this is put forward as a novel mechanism the manuscript lacks detail that would provide confidence to the reader that these findings are indeed as exciting as proposed.

Considerations, questions and comments:

Clearly the researchers have established the methods for rHPLC separation of insulin products generated under various redox conditions (Fig 1). However, no data is provided to demonstrate the definitive mass spec identification of these products. Spectra demonstrating this should be included in the supplementary data, allowing the reader to connect specific HPLC peaks to a definitively identified product arising from thiol-disulphide exchange. Particularly important would be to provide spectra of the A and B chain products with the additional intrachain disulphide included. As all of the assumptions rely on this identification this information underpins the entire manuscript and hypothesis. It would also be good to provide evidence for any intermediates/partially reduced forms (as shown in Fig. 1b). In addition, it would be good to include a time course of human insulin degradation as per Fig 3 and Fig 4 in the supplementary file (even though this is likely to demonstrate a much more rapid degradation profile).

Is it possible to get human insulin chain splitting as a result of the ionisation conditions used during mass spectrometry? Do you ever see the two A and B chain with or without intrachain disulphides arising spontaneously during mass spec analysis? It would be good to definitively show this.

No circular dichroism spectra are provided and should be included in the supplementary section. This will enable assessment of the unfolding data. The raw spectra will also provide an indication of whether the mutations are affecting the overall secondary structure and not just the unfolding. It would be good to know if any of the spectra show a decrease in secondary structure prior to the unfolding experiment. Currently the reader is being asked to trust that the CD spectra are all equal at the beginning.

Interestingly, three molecules of icodec are found in the asymmetric subunit of the crystal structure. Two molecules are in a dimeric arrangement essentially the same as dimeric insulin previously described. The third molecule has structural differences at either end of the B chain. On page 7 line 178 it is stated that this arrangement "is consistent with self-association properties different from those of human insulin". Where is this evidence? Does it appear in other publications and if so please provide the reference.

Page 9 line 235 "indicate that the undesired degradation via insulin chain-splitting is highest in rats..." Fig 4 does not provide evidence for "chain-splitting" but merely shows the remaining intact insulin, which could be due to a number of other modifications, although the authors have discounted proteolytic cleavage. The kinetics of appearance of "split" chains in Fig 4 should be provided.

How can you explain that the in vitro plasma stability for pig serum is much greater than the stability in vivo in minipigs? This doesn't seem to be the case for the human plasma studies in vitro versus in vivo.

The authors place emphasis on the lack of proteolytic degradation detectable in Fig 3-5 being

evidence that the chain-splitting is the main mechanism for degradation. How then do they reconcile the fact that proteolytic products are found in the human experiments (Fig 6)? Is this pointing to lack of sensitivity of proteolytic product detection in the previous experiments in Fig 3-5?

Page 11 Line 291: "Given the lack of support of proteolysis in serum as an insulin-degradation mechanism"??

It would be helpful to include a table in the supplementary section defining all of the analogues used. Particularly, "C18" and "C20" have not been well defined here. Even though they have been described elsewhere it would be a lot easier to read if this information was provided.

Fig. 1. Legend should include definition of colouring in Fig 1a, light blue, dark blue, ? yellow and ? grey. It is not really clear whether the the A7-B7 bond is grey.

The order of Fig. 1 could be changed to have the HPLC spectra prior to presenting the data currently in Fig. 1c, so that the conditions for identification of % intact are clarified first.

Fig. 1c the pink line label has a typo A14E, deB30 - ? should be A14E, desB30.

The panels in Fig. 1d (HPLC spectra) need to be labelled to identify the redox conditions used in each panel. The different A chain products presumably have been characterised and should be appropriately labelled to indicate what they are. Which peak represents the B-chain with additional intrachain disulphide? This needs to be identified and labelled.

Fig. 2. This figure would benefit from some labels to show the numbering/identity of the amino acids shown. It would also be good to include the residues A14, B16, B25 on Fig. 2c left

Fig. 2c shows dotted lines depicting interactions. The distances are shown but are barely readable. Can you comment on the fact that the distance is shorter between A17 and B22 in human insulin versus icodec – this seems contrary to the statement that the electrostatic interaction with Arg B22 is greater for icodec.

Fig. 3 and Fig 4. It would be good to retain the same colouring (cyan) and symbols (downward triangle) for icodec here as compared to Fig 1. In fig 3 this scheme is applied to A14E C18.

Fig 3b. Whilst the legend mentions all containing the internal disulphide bond it is not clear exactly which species of "B-chains" were identified in Fig 3b to generate the "MS response". It would be good to clarify this, especially as there is a claim on page 8 line 224 that "a lack of other B-chain degradation products provides unequivocal support for the hypothesis". What about the A chain product? Do you have evidence that this also follows a similar pattern?

Fig 5. What mass species of idodec B-chain with intrachain disulphide is being detected and plotted?

REVIEWER COMMENTS

Reviewer #1 (Remarks to the Author):

This is an excellent work reported by the authors about an underappreciated mechanism for insulin clearance in vivo. This is indeed a special circumstance as insulin normally gets cleared way before the disulfide shuffling can happen in blood. I thought this study is clearly presented with convincing evidence to support the conclusion. I am surprised and satisfied to see that the different mutations and fatty acid attachment can lead to different degrees of stabilization in GnHCl condition or redox condition.

I really do not have anything to add and would be happy to see this work published as is.

On the other hand, while I think this may be out of the scope of this work, the authors may want to explore insulin analogs where the disulfide bonds are replaced with non-reducing bonds such as thioether bonds. Will be informative to see if the existence of insulin B chain is still present.

Although outside of the scope of the present manuscript, we agree with the reviewer that it would be interesting to investigate insulin analogues replacing the disulfide bonds with non-reducing bonds. This could be a potential future field of research.

Reviewer #2 (Remarks to the Author):

This paper reports a new insulin clearance mechanism, i.e., splitting insulin into its A-chain and B-chain by a thiol–disulphide exchange reaction. Insulin icodec, an insulin analogue in clinical development for once-weekly dosing, seems to be protected against this clearance mechanism, although not completely, which contributes to its long half-life.

Identification of this new clearance mechanism may have a significant scientific impact, particularly for the development of insulin analogues with a long half-life.

The paper is well written and the results are convincing.

This reviewer only has minor comments:

1. Page 2, last sentence: The impact of insulin icodec on glucose control (HbA1c) has not been consistent in the phase 3 programme. ONWARDS 6 reports less improvement in HbA1c with icodec vs degludec in people with type 1 diabetes. It might be the best to delete this sentence as the glucose-lowering effect of icodec is of little relevance for the purpose of this paper.

We agree with the reviewer and have deleted this sentence as suggested.

2. Methods: It would be good to learn more about the sensitivity (detection limits) of the LC-MS for the different metabolites.

Lower limits of quantification for the reported metabolites were included in the methods section.

3. The authors were able to detect free icodec B-chains in the plasma of mini-pigs, whereas free A-chains were not seen. The authors explain this with renal clearance of free A-chains (renal clearance of B-chains is

supposed to be limited). It would be great to see some supporting data with A-chain and B-chain measurements in urine. Or are concentrations too low (and the assay too insensitive) to detect the chains in urine.

We thank the reviewer for pointing this out, and we have updated the text in the manuscript to include “and/or proteolytic degradation”. We can only see that it disappears fast, which could either be due to renal clearance or due to proteolytic degradation.

4. The authors show that “the rate-limiting step of the thiol–disulphide-based reaction is the free thiol (glutathione) attack of a disulphide bond”. May this be of clinical importance, e.g. in people taking glutathione supplements?

This is a very interesting question. Indeed, any changes in redox environment in plasma will likely change insulin stability. Such changes could be brought about by pathophysiological processes such as obesity or by different diets. For example, it was previously reported that Mediterranean diet influences plasma redox profile independent of BMI in healthy adults (Higher Mediterranean Diet Quality Scores and Lower Body Mass Index Are Associated with a Less-Oxidized Plasma Glutathione and Cysteine Redox Status in Adults). This topic is definitely of interest for further studies.

5. The authors understandably focus on the importance of the thiol based degradation pathway for long-acting insulin analogues, as the degradation rates are too slow for insulins with a short half-life. What about intermediate-acting analogues such as detemir? Can the reduced potency of detemir be partly explained by a higher thiol based degradation pathway compared to other analogues?

The reviewer brings up a very interesting point. Indeed, we are currently working on a different manuscript addressing the stability of disulfide bonds in insulin Detemir. We replaced a sentence in the discussion section to highlight that both the disulfide bond stability of the individual molecule and its circulating half-life are important to consider.

Reviewer #3 (Remarks to the Author):

This manuscript describes the proposed identification of a novel mechanism by which insulin is degraded. Discussed are the implications for regulation of circulating insulin and long acting insulin analogues via this degradation mechanism. It is argued that whilst this mechanism likely only contributes in a minor way to regulation of insulin it has potentially significant impact on circulating long acting analogues (using icodec as the model in this case).

The manuscript provides evidence for the existence of “split” A and B chains generated through thiol–disulphide exchange, leading to formation of new intrachain disulphide bonds in both chains. Such a mechanism for insulin degradation is truly novel and has important implications for future long-acting insulin analogue design. Whilst this is put forward as a novel mechanism the manuscript lacks detail that would provide confidence to the reader that these findings are indeed as exciting as proposed.

Considerations, questions and comments:

Clearly the researchers have established the methods for rHPLC separation of insulin products generated under various redox conditions (Fig 1). However, no data is provided to demonstrate the definitive mass spec identification of these products. Spectra demonstrating this should be included in the supplementary data, allowing the reader to connect specific HPLC peaks to a definitively identified product arising from thiol–disulphide exchange. Particularly important would be to provide spectra of the A and B chain products with the additional intrachain disulphide included. As all of the assumptions rely on this identification this

information underpins the entire manuscript and hypothesis. It would also be good to provide evidence for any intermediates/partially reduced forms (as shown in Fig. 1b).

Thank you for pointing this out. MS spectra for insulin degradation products identified under different redox conditions have been included in supplementary information.

In addition, it would be good to include a time course of human insulin degradation as per Fig 3 and Fig 4 in the supplementary file (even though this is likely to demonstrate a much more rapid degradation profile).

We agree with the reviewer that a comparative time course of human insulin in plasma is beneficial to highlight, how fast intact human insulin disappears and that the cause of this is due to extensive proteolytic degradation, with no free B-chain being detected by incubation in rat plasma, which is in sharp contrast to the results observed with all the fatty acid acylated insulin variants. We have included this as a supporting figure (supporting figure 4) and have added the following text to the manuscript:

“The detection of the free B-chain, and the fact that we were unable to identify any proteolytic degradation products in these incubations, is in contrast to incubating human insulin in rat plasma, which rapidly disappears and only proteolytic degradation products and no free B-chain being detected (Supporting Figure 4).”

Is it possible to get human insulin chain splitting as a result of the ionisation conditions used during mass spectrometry? Do you ever see the two A and B chain with or without intrachain disulphides arising spontaneously during mass spec analysis? It would be good to definitively show this.

We thank the reviewer for the insightful question. It is indeed possible to reduce disulfide bonds during the electrospray ionization mechanism under certain conditions and on certain commercially available instruments. This is a field of research we have been active in (Cramer, C. N., et al. (2017). "Complete Mapping of Complex Disulfide Patterns with Closely-Spaced Cysteines by In-Source Reduction and Data-Dependent Mass Spectrometry." *Anal Chem* 89(11): 5949-5957. and Cramer, C. N., et al. (2018). "Generic Workflow for Mapping of Complex Disulfide Bonds Using In-Source Reduction and Extracted Ion Chromatograms from Data-Dependent Mass Spectrometry." *Anal Chem* 90(13): 8202-8210.). However, the in-source reduction efficiency is very poor on peptides connected by 2 or more disulfide bonds such as the insulin molecule (<1%), even at optimized conditions for induction of in-source reduction of disulfide bonds. In LC-MS, disulfide reduced and/or chain splitting products will typically have retention times different from the parent molecule as is the case in our analyses. However, in-source reduction should be suspected if perfect co-elution profiles are observed between such products and the parent molecule.

No circular dichroism spectra are provided and should be included in the supplementary section. This will enable assessment of the unfolding data. The raw spectra will also provide an indication of whether the mutations are affecting the overall secondary structure and not just the unfolding. It would be good to know if any of the spectra show a decrease in secondary structure prior to the unfolding experiment. Currently the reader is being asked to trust that the CD spectra are all equal at the beginning.

The raw spectra were included as supplementary Figure 3.

Interestingly, three molecules of icodec are found in the asymmetric subunit of the crystal structure. Two molecules are in a dimeric arrangement essentially the same as dimeric insulin previously described. The third molecule has structural differences at either end of the B chain. On page 7 line 178 it is stated that this arrangement “is consistent with self-association properties different from those of human insulin”. Where is this evidence? Does it appear in other publications and if so please provide the reference.

Thanks for pointing this out. This sentence has been deleted from the manuscript as detailed discussion of insulin icodec self-association properties is outside of the scope of this manuscript and may be subject of future publication.

Page 9 line 235 “indicate that the undesired degradation via insulin chain-splitting is highest in rats...” Fig 4 does not provide evidence for “chain-splitting” but merely shows the remaining intact insulin, which could be due to a number of other modifications, although the authors have discounted proteolytic cleavage. The kinetics of appearance of “split” chains in Fig 4 should be provided.

The corresponding appearance of B-chains relating to the manuscript figure 4 has now been included as a supporting figure (supporting figure 5) and a reference made in the revised manuscript figure 4 figure legend.

How can you explain that the *in vitro* plasma stability for pig serum is much greater than the stability *in vivo* in minipigs? This doesn't seem to be the case for the human plasma studies *in vitro* versus *in vivo*.

Thank you for pointing this out. We have no simple explanation. It is inherently difficult to compare the results from *in vitro* incubations and *in vivo* studies in general and even more difficult to compare across different species. There are several differences that might influence the results: the minipig study is based on acute *i.v.* administration compared to the multiple *s.c.* dosing in humans. Also, the half-life of insulin icodex in minipigs (50 h) is shorter than its half-life in humans (192 h), and we do not know the half-life of the B-chain in either species. We also do not know whether degradation *in vivo* is limited to the plasma compartment, so the rates of degradation *in vitro* and *in vivo* may be both different and species dependent.

The authors place emphasis on the lack of proteolytic degradation detectable in Fig 3-5 being evidence that the chain-splitting is the main mechanism for degradation. How then do they reconcile the fact that proteolytic products are found in the human experiments (Fig 6)? Is this pointing to lack of sensitivity of proteolytic product detection in the previous experiments in Fig 3-5?

We thank the reviewer for the question. In figure 6 the steady state profile of insulin icodex following the 5th weekly *s.c.* dose in humans is showed, as well as the profile of the free B-chain and two additional metabolites. We believe that the reason why we were able to detect the two metabolites from proteolytic degradation in these samples are two-fold: first, this is because of the high overall exposure following accumulation due to the multiple doses (i.e. five). And second, the B24-29 and B29 metabolites most likely predominantly are formed from the free B-chain following chain-splitting, thus being a down-stream product following chain-splitting.

The two most obvious reasons for not being able to detect these proteolytic products in the *i.v.* minipig study is that they either are not formed, or that the exposure of such products is below our detection limit in this much lower overall exposure study.

The strongest arguments for chain-splitting being the main degradation mechanism is that the free B-chain was the only degradation product observed in the minipig study, and that it was the dominant degradation product in humans, even though that B-chain most likely is an intermediate in the formation of the lower abundant B24-B29 and B29 metabolites. We have adjusted the thickness of the different lines in Figure 6b to make it clearer.

Page 11 Line 291: “Given the lack of support of proteolysis in serum as an insulin-degradation mechanism”?

Thank you for pointing this out. We re-phrased this paragraph in the discussion to reflect that the levels of proteolytic degradation products for fatty-acid protracted insulin analogs are minor in comparison to the degradation products originating from the re-arrangements of disulphide bonds.

It would be helpful to include a table in the supplementary section defining all of the analogues used. Particularly, “C18” and “C20” have not been well defined here. Even though they have been described elsewhere it would be a lot easier to read if this information was provided.

Section describing structures of the compounds was included in supplementary information.

Fig. 1. Legend should include definition of colouring in Fig 1a, light blue, dark blue, ? yellow and ? grey. It is not really clear whether the the A7-B7 bond is grey.

Colouring definitions were added to Figure 1 legend.

The order of Fig. 1 could be changed to have the HPLC spectra prior to presenting the data currently in Fig. 1c, so that the conditions for identification of % intact are clarified first.

Figure 1 was reordered according to the reviewer's suggestion.

Fig. 1c the pink line label has a typo A14E, deB30 - ? should be A14E, desB30.

The typo has been corrected.

The panels in Fig. 1d (HPLC spectra) need to be labelled to identify the redox conditions used in each panel. The different A chain products presumably have been characterised and should be appropriately labelled to indicate what they are. Which peak represents the B-chain with additional intrachain disulphide? This needs to be identified and labelled.

The panel in Figure 1d (which now is Figure 1c) labelling was improved according to the suggestions and additional MS results incorporated into the supplementary Figure 2.

Fig. 2. This figure would benefit from some labels to show the numbering/identity of the amino acids shown. It would also be good to include the residues A14, B16, B25 on Fig. 2c left.

Labels of all depicted side chains are now included in the figure.

Fig. 2c shows dotted lines depicting interactions. The distances are shown but are barely readable. Can you comment on the fact that the distance is shorter between A17 and B22 in human insulin versus icodec – this seems contrary to the statement that the electrostatic interaction with Arg B22 is greater for icodec.

The font size has been increased so that distances are now easier to read. As insulin icodec crystallized at pH 4.6 glutamates might be protonated explaining the longer distances. Most importantly the location between the negatively charged amino acids and the C-terminus of the A chain has been retained. This has now been added to the text and a Figure of an insulin with the same backbone mutations as icodec has been added to Fig 2c for comparison.

Fig. 3 and Fig 4. It would be good to retain the same colouring (cyan) and symbols (downward triangle) for icodec here as compared to Fig 1. In fig 3 this scheme is applied to A14E C18.

We agree, and thank the reviewer for pointing this out. We have updated the figures in the revised manuscript incl more consistent choices of colors and symbols across figures. In addition to this, we have corrected a mistake in figure 3 and 4, where the numbering of the y-axes was incorrect.

Fig 3b. Whilst the legend mentions all containing the internal disulphide bond it is not clear exactly which species of "B-chains" were identified in Fig 3b to generate the "MS response". It would be good to clarify this, especially as there is a claim on page 8 line 224 that "a lack of other B-chain degradation products provides unequivocal support for the hypothesis". What about the A chain product? Do you have evidence that this also follows a similar pattern?

We have updated in the figure legends the following "All insulin B-chains contain an internal disulphide bond connecting Cys 7 with Cys 19, **assessed by the exact monoisotopic masses**". This was possible, since we used high-resolution mass spectrometers for the analyses.

Fig 5. What mass species of icodec B-chain with intrachain disulphide is being detected and plotted?

We thank the reviewer for allowing us to clarify this. The samples were measured up against standard curves of both intact insulin icodec and the B-chain containing the internal disulfide bond. We have clarified this by updating the following sentence in the methods section: "For quantification of the in vivo minipig samples, selected plasma standards with either intact insulin icodec or the insulin icodec B-chain reference standard with an internal disulfide bond connecting Cys 7 with Cys 19 at concentrations in the range 0.05–200 nM were prepared"

Reviewers' Comments:

Reviewer #2:

Remarks to the Author:

All my comments were fully addressed. This is a very interesting paper offering new insights into insulin degradation.

Reviewer #3:

Remarks to the Author:

The authors have addressed this reviewer's comments and I have no further comments.